# Western-Type *Helicobacter pylori* CagA are the Most Frequent Type in Mongolian Patients

**DOI:** 10.3390/cancers11050725

**Published:** 2019-05-24

**Authors:** Tegshee Tserentogtokh, Boldbaatar Gantuya, Phawinee Subsomwong, Khasag Oyuntsetseg, Dashdorj Bolor, Yansan Erdene-Ochir, Dashdorj Azzaya, Duger Davaadorj, Tomohisa Uchida, Takeshi Matsuhisa, Yoshio Yamaoka

**Affiliations:** 1Department of Gastroenterology, Mongolian National University of Medical Sciences, Ulaanbaatar city 14210, Mongolia; tserentogtoht@yahoo.com (T.T.); medication_bg@yahoo.com (B.G.); oyuntsetseg.kh@mnums.edu.mn (K.O.); davaadorj55@hotmail.com (D.D.); 2Department of Endoscopy, Medipas hospital, Orkhon province, Bayan-Undur soum, Zest bag 61029, Mongolia; 3Department of Environmental and Preventive Medicine, Oita University Faculty of Medicine, Yufu city 879-5593, Japan; phawinee@oita-u.ac.jp (P.S.); m16d9102@oita-u.ac.jp (D.A.); 4Department of Endoscopy, National Cancer Center, Ulaanbaatar city 13370, Mongolia; boogii6655@gmail.com; 5Department of General Surgery National Cancer Center, Ulaanbaatar city 13370, Mongolia; erdeneochiry@gmail.com; 6Department of Molecular Pathology, Oita University Faculty of Medicine, Yufu city 879-5593, Japan; tomohisa@oita-u.ac.jp; 7Department of Endoscopy, Nippon Medical University Tama Nagayama Hospital, Tokyo 113-8602, Japan; matuhisa@m8.dion.ne.jp; 8Global Oita Medical Advanced Research Center for Health, Yufu city 879-5593, Japan; 9Department of Medicine, Gastroenterology and Hepatology section, Baylor College of Medicine, Houston, TX 77030, USA

**Keywords:** *Helicobacter pylori*, CagA, VacA, genotyping, gastric cancer, Mongolia

## Abstract

*Helicobacter pylori* infection possessing East-Asian-type CagA is associated with carcinogenesis. Mongolia has the highest mortality rate from gastric cancer. Therefore, we evaluated the CagA status in the Mongolian population. High risk and gastric cancer patients were determined using endoscopy and histological examination. *H. pylori* strains were isolated from different locations in Mongolia. The CagA subtypes (East-Asian-type or Western-type, based on sequencing of Glu-Pro-Ile-Tyr-Ala (EPIYA) segments) and *vacA* genotypes (s and m regions) were determined using PCR-based sequencing and PCR, respectively. In total, 368 patients were examined (341 gastritis, 10 peptic ulcer, and 17 gastric cancer). Sixty-two (16.8%) strains were *cagA*-negative and 306 (83.1%) were *cagA*-positive (293 Western-type, 12 East-Asian-type, and one hybrid type). All *cagA*-negative strains were isolated from gastritis patients. In the gastritis group, 78.6% (268/341) had Western-type CagA, 2.9% (10/341) had East-Asian-type, and 18.2% (61/341) were *cagA*-negative. However, all *H. pylori* from gastric cancer patients possessed Western-type CagA. Histological analyses showed that East-Asian-type CagA was the most virulent strains, followed by Western-type and *cagA*-negative strains. This finding agreed with the current consensus. *CagA*-positive strains were the most virulent type. However, the fact that different CagA types can explain the high incidence of gastric cancer might be inapplicable in Mongolia.

## 1. Introduction

*Helicobacter pylori* infection causes chronic gastritis, inducing chronic inflammation in the gastric mucosa, depending on the pattern of inflammation throughout the stomach. It can cause peptic ulcer diseases (PUDs) or lead to the development of premalignant lesions. The development of gastric cancer occurs in the context of chronic atrophic gastritis, which is a consequence of the infection. Approximately 90% of gastric cancers, especially non-cardiac cancers, are thought to occur due to *H. pylori* infection [1]. Approximately 70% of the world’s gastric cancers occur in East Asia; the top four countries by incidence in either sex are South Korea (age standardized rate per 100,000 people: 39.6), followed by Mongolia (33.1), Japan (27.5), and China (20.7) (GLOBOCAN2018; http://globocan.iarc.fr/).

We previously hypothesized that differences in gastric cancer incidence among geographic regions can be partially explained by differences in *H. pylori* virulence factors, especially CagA and VacA [2]. The major *H. pylori* virulence factor CagA is thought to play an important role in gastric carcinogenesis [2,3,4,5,6]. There are two types of clinical isolates: CagA-producing (*cagA*-positive) and CagA non-producing (*cagA*-negative) strains. Almost all *H. pylori* isolates from East Asia are *cagA*-positive, and approximately 20–40% of isolates from Europe and Africa are *cagA*-negative. Different numbers of repeat sequences are located in the 3′-region of *cagA* [7], which contains the Glu-Pro-Ile-Tyr-Ala (EPIYA) motif. The sequences are annotated according to segments of 20–50 amino acids flanking the EPIYA motifs (segments EPIYA-A, -B, -C, or -D, in which EPIYA-C is specific for Western-type CagA and EPIYA-D is specific for East-Asian-type CagA) [2,4]. The current consensus is that East-Asian-type CagA is associated with gastric carcinoma.

VacA is another extensively studied *H. pylori* virulence factor. Differences in *vacA* structure at the signal (s) region (s1 and s2) and the middle (m) region (m1 and m2) [8] are associated with variation in vacuolating activity among *H. pylori* strains [9]. Based on in vitro experiments, s1m1 strains are the most cytotoxic, followed by s1m2 strains and s2m2 strains, which have no cytotoxic activity [8]. In agreement with in vitro data, many studies in Western countries showed that individuals infected with *vacA* s1 or m1 *H. pylori* strains have a higher risk of PUD than those infected with s2 or m2 strains [8,10]. Importantly, the presence of CagA and VacA is typically linked. Hence, *H. pylori* either produces both or neither of the proteins [2].

Mongolia is in East-Central Asia. It borders Russia to the north and China to the south, east and west. Mongolia is the second highest country by gastric cancer incidence in the world and the country with the highest gastric cancer mortality (http://globocan.iarc.fr/). Our previous study showed that 75.9–80% of Mongolian dyspeptic patients were infected with *H. pylori* [11,12]. Importantly, our previous immunohistochemistry data from one location, Ulaanbaatar City, Mongolia, showed that specific antibodies for East-Asian-type CagA could not react in most cases (95–99.4%) with anti-CagA polyclonal antibody positive cases. Among *H. pylori* positive cases, 18.8% did not react with polyclonal anti-CagA antibody [13], suggesting the presence of *cagA*-negative strains are seemed to exist among Mongolian population. Therefore, we constructed a nation-wide study to evaluate the CagA status directly from cultured *H. pylori* using sequencing in Mongolian patients including gastric cancer patients.

## 2. Results

### 2.1. Patient Demographics

We initially enrolled 1004 patients, and we successfully isolated *H. pylori* from 404 patients. Thirty-six cases were excluded due to a lack of histology or insufficient biopsy specimens, so we used 368 cases for further analysis. Among them, 21 cases were from Uvs Province, 31 cases from Khuvsgul province, 89 from Khentii province, 93 from Umnugovi province and 116 cases from Ulaanbaatar city. We obtained 16 cases with gastric cancer from the National Cancer Center and one gastric cancer case from an Ulaanbaatar survey. Overall, 341 cases were gastritis, 10 cases were peptic ulcers (seven cases with gastric ulcer and three cases with duodenal ulcer) and 17 cases were gastric cancer patients. 70.1% (258/368) were female and 29.9% (110/368) were male. The ages ranged from 16 to 81 years old, and the mean age with standard deviation was 43.1 ± 14 years. Detailed patient information related to the clinical setting is shown in Appendix A.

### 2.2. CagA Genotypes

The distributions of the CagA types are shown in Table 1. We found that only 83.2% (306/368) of strains possessed the *cagA* gene, which was far lower than in other East-Asian countries where it is nearly 100% [2]. The remaining 16.8% (62/368) was *cagA*-negative. More surprisingly, 95.8% (293/306) of *cagA*-positive strains were Western-type (e.g., ABC, ABCC), only 3.9% (12/306) were East-Asian type (nine with ABD and three with ABBD), and 0.3% (1/306) were mixed East-Asian/Western-type strain (ABDC (hybrid type)). ABC and ABCC subtypes were the most common at 41.6% and 31.0%, respectively. 

### 2.3. CagA and vacA Genotyping Based on Diseases

Figure 1A shows the distribution of the CagA type based on the disease group. Importantly, all *cagA*-negative strains were isolated from gastritis patients (22.2% (62/279)), and all strains from gastric cancer patients were Western-type CagA (100% (17/17)) (*p* < 0.03). In contrast, there was no significant relationship among CagA patterns, clinical outcomes, and the geographic location (Appendix A).

The predominant *vacA* genotype was s1 (83.2% (306/368)), and the other 62 strains (16.8%) were the s2 genotype. The *vacA* m1 genotype was found in 56.5% (208/368) and *vacA* m2 was found in 43.5% (160/368). Based on the combination of the s and m regions, s1m1 was most common (204/368, 55.4%), followed by s1m2 (102/368, 27.7%) and s2m2 (62/368, 16.8%). All s2 strains were m2 type and *cagA*-negative. The s2 genotype was only found in strains from gastritis patients that resembled *cagA*-negative strains (Figure 1B), and the gastritis vs. gastric cancer comparison was statistically significant (*p* < 0.04). 

The prevalence of the *vacA* m1 genotype was the highest in strains from gastric cancer (76.5% (13/17)), followed by peptic ulcer (60% (6/10)) and gastritis (55.4% (189/341)) patients, but the differences were not statistically significant (Figure 1C). There was no significant relationship among *vacA* genotypes patterns, clinical outcomes, and the geographic location (Appendix A).

The most predominant genotype in the *cagA* and *vacA* gene combination was *cagA*-positive, *vacA* s1/m1 (55.4% (204/368)), followed by *cagA*-positive, *vacA* s1/m2 (27.7% (102/368)) and *cagA*-negative, *vacA* s2/m2 (16.8% (62/3680)). The *cagA*-negative/*vacA* s2/m2 strains were only observed in strains from gastritis patients (18.2% (61/341)). The prevalence of the *cagA*-positive/*vacA* s1/m1 genotype was the highest in strains from gastric cancer patients (76.5% (13/17)), followed by those from peptic ulcer patients (60% (6/10)) and those from gastritis patients (54.3% (185/341)), but the differences were not statistically significant (Figure 1D).

Next, we focused on Western-type CagA typing (EPIYA AB (AB, ABB, and ABBB), ABC (ABC and ABBC), ABC* motif (more than one EPIYA C repeats: ABCC and ABCCC motif)) for diseases (Figure 2). There was no statistical significance for Western-type CagA subtyping by disease groups. 

### 2.4. Gastric Mucosal Status with Respect to H. pylori Virulence Factors

The gastric mucosal status, based on neutrophil and monocyte infiltration, atrophy, and intestinal metaplasia, was evaluated according to the updated Sydney system [14] in non-gastric cancer patients. First, we compared *cagA*-positive (*vacA* s1) to *cagA*-negative (*vacA* s2). All histology scores were higher in the *cagA*-positive group than the *cagA*-negative group in each of the three locations, especially for angulus and corpus (Appendix A). The activity (neutrophil infiltration) score was significantly higher in *cagA*-positive (*vacA* s1) than in *cagA*-negative (*vacA* s2) cases (mean (median); 0.7 (1) vs. 1.1 (1), *p* = 0.0001 for angulus, 0.5 (0.5) vs. 0.8 (1), *p* = 0.007 for corpus). The inflammation (monocyte infiltration) score in the antrum, angulus, and corpus was also significantly higher in *cagA*-positive (*vacA* s1) than in *cagA*-negative (*vacA* s2) cases: 1.1 (1) vs. 1.2 (1), *p* = 0.05, 1.2 (1) vs. 1.6 (2), *p* = 0.006 and 0.6 (0.5) vs. 0.9 (1), *p* = 0.04, respectively. The atrophy score was significantly higher in *cagA*-positive cases than *cagA*-negative cases in the angulus (0.6 (0) vs. 1 (1), *p* = 0.04) and in the corpus 0.1 (0) vs. 0.3 (0), *p* = 0.05).

We focused on the CagA types among *cagA*-positive cases (Figure 3). Patients infected with East-Asian type CagA strains had significantly higher scores of intestinal metaplasia (IM) in the antrum (0.4 (0)) and angulus (0.7 (0)) than those with Western-type CagA strains (0.09 (0); *p* = 0.01 and 0.2 (0) *p* = 0.01, respectively). Next, we focused on Western-type CagA typing (EPIYA AB (AB, ABB, and ABBB), ABC (ABC and ABBC), ABC* motif (more than one EPIYA C repeats: ABCC and ABCCC motif)) (Appendix A). Histology scores were generally higher in multiple ABC* motif cases than in cases of ABC followed by the AB motif. In the angulus, the activity score was significantly higher in cases with multiple ABC* than ABC followed by the AB motif: 1.2 (1), 1 (1) and 0.8 (1) respectively (*p* <0.0001 for each). In the corpus, the activity score was significantly higher in cases with ABC than cases with the AB motif (0.8 (1) and 0.6 (0) *p* = 0.001).

We focused on the *vacA* m genotypes among *cagA*-positive cases (i.e., s1 genotypes). As expected, all histology scores were generally higher in *vacA* s1m1 than in the *vacA* s1m2 subtype (Figure 4). In the angulus, *vacA* s1m1 cases showed significantly higher atrophy and IM scores than s1m2 cases (1.0 (1) vs. 0.8 (1), *p* = 0.05 for atrophy and 0.3 (0) vs. 0.1 (0), *p* = 0.04 for IM). In addition, neutrophil infiltration, monocyte infiltration and atrophy scores were also significantly higher in *vacA* s1m1 cases than in s1m2 cases (0.9 (1) vs. 0.7 (1), *p* = 0.01 for neutrophil infiltration; 1.0 (1) vs. 0.6 (1), *p* = 0.0001 for monocyte infiltration, and 0.3 (0) vs. 0.1 (0), *p* = 0.01 for atrophy). 

Multivariate logistical regression analysis showed that the East-Asian-type CagA was associated with a higher risk for IM, followed by Western-type EPIYA-ABCCC, ABCC, ABC and AB motif compared to *cagA*-negative strains. The ABCC motif was associated with a significantly higher risk for neutrophil infiltration and atrophy. Detailed odds ratios and statistical significance are summarized in Appendix A. We examined the combination of the *vacA* genotype. The *vacA* s1/m1 genotype was associated with a higher risk for neutrophil infiltration, atrophy, and IM, followed by *vacA* s1/m2, compared to the *vacA* s2/m2 genotype. Detailed odds ratios and statistical significance are summarized in Appendix A.

## 3. Discussion

The majority of gastric cancer cases originate in East-Asian countries where *H. pylori* carrying East-Asian-type CagA are thought to be responsible for carcinogenesis [3]. This consensus is based on several previous reports as follows [4,15,16]. East-Asian-type CagA containing EPIYA-D segments exhibits a stronger binding affinity for the src homology-2 domain-containing phosphatase 2 (SHP2) and a greater ability to induce morphological changes in epithelial cells than Western-type *cagA* containing EPIYA C segments [4]. Additionally, transgenic mice expressing East-Asian-type CagA develop gastrointestinal and hematopoietic malignancies more frequently than mice expressing Western-type CagA [15]. There are also many previous epidemiological studies showing that East-Asian-type CagA is a significant risk factor for gastric cancer compared to Western-type *cagA*, including our studies in Okinawa, the south islands of Japan [16].

Since Mongolia is in East Asia and it is the country with the second highest incidence of gastric cancer, we hypothesized that *H. pylori* strains in Mongolia should contain East-Asian-type CagA. Surprisingly, all Mongolian patients with gastric cancer were infected with Western-type CagA (Figure 1A). Moreover, although Western-type CagA *H. pylori* possessing more than one EPIYA C repeat is reported to be associated with a higher risk for gastric carcinogenesis [4,17,18], the majority of gastric cancer patients were infected with *H. pylori* possessing a single EPIYA C repeat (Table 1). Although our study was limited by the small number of gastric cancer cases, we clearly showed that the current consensus that strains containing East-Asian-type CagA or multiple repeat regions of the EPIYA C segment are not sufficient virulent markers to explain gastric cancer development. 

However, we also found that gastric mucosal damage, especially IM scores, were higher in patients infected with East-Asian-type CagA strains than those with Western-type CagA strains (Figure 3). Moreover, patients infected with *H. pylori* possessing multiple EPIYA C repeats usually had higher histological features of gastritis than those with *H. pylori* possessing a single or no EPIYA C repeats (Appendix A). Therefore, our data support the consensus that East-Asian-type CagA or Western-type CagA strains with multiple EPIYA C repeats are the virulent forms of CagA that cause gastric mucosal damage based on the presence of IM, the precancer status. We previously compared gastric the mucosal background between Japanese and Mongolian dyspeptic patients and found that all histological features of gastritis (activity, inflammation, atrophy and IM) were significantly higher in Japanese patients [12]. This could be explained by CagA typing showing. In this case, Mongolian dyspeptic patients were infected with less virulent strains (Western-type CagA) and East-Asian-type CagA would be the marker for the severe histological features of gastritis. In addition, we found that the presence of *cagA*-positive *H. pylori* was associated with the presence of gastric precursor diseases and gastric cancer, supporting the consensus that the presence of *cagA* increased gastric cancer risk [19,20,21]. 

However, our study clearly showed that there is a gap between gastric mucosal damage and the development of gastric cancer. East-Asian-type CagA is not a sufficient marker for the presence of gastric cancer. Overall, most Mongolian dyspeptic patients are infected with less virulent strains like Western-type CagA. Therefore, the gastric mucosal damages are not severe, but factors other than CagA types must be involved in the development of gastric cancer.

Other *H. pylori* virulence factors could contribute to increased virulence. Supporting the consensus that the *vacA* s1/m1 genotype is more virulent than the s1/m2 genotype [22], gastric mucosal damage was higher in patients infected with s1/m1 strains than those with s1/m2 strains (Figure 4). The prevalence of s1/m1 strains was also the highest in gastric cancer patients (Figure 1D), suggesting a role for *vacA* genotypes in the development of gastric cancer. In addition, all *cagA*-positive strains were of the *vacA* s1 genotype, supporting the consensus that the *vacA* s1 genotype is linked with *cagA*-positive status [2]. Since virulence factors are linked, studies on groups of putative virulence factors are needed to provide clinically useful information [23]. *H. pylori* contains approximately 1600 genes, and it is likely that only a fraction of the potential virulence genes have been identified. Therefore, additional whole genome analysis of *H. pylori* is required to screen other important virulence factors in a larger number of gastric cancer cases.

In addition to *H. pylori* factors, other pathogenic infections and changes to the host resident microbiome might explain gastric cancer development in the Mongolian population. This possibility is currently being investigated. Apart from bacterial infection, our previous study showed that environmental factors like a high salt diet, hot beverages, tobacco smoking and a diet poor in fresh fruits were associated with incidence of gastric cancer in the Mongolian population [13].

Finally, we previously reported that the majority of gastric cancers in Mongolia were located in the upper part of stomach [12,13] but most gastric cancer arises in the lower part of the stomach in other countries with high incidences of gastric cancer like Japan, Korea and China [24]. Gastric atrophy and IM are thought to be the precursor diseases for gastric cancer development [25]. According to our previous study, in addition to atrophy or IM, corpus predominant gastritis was associated with a higher risk for gastric cancer in the Mongolian population [26]. Therefore, the mechanisms contributing to the development of gastric cancer in Mongolia might differ from those in other countries with high incidence of gastric cancer where *H. pylori*, especially CagA types, play important roles. 

## 4. Materials and Methods

### 4.1. Study Population and Sampling

We conducted a cross sectional study among gastric cancer and non-gastric cancer patients, from November 2014 to August 2016. Dyspeptic patients were recruited in Ulaanbaatar City (18–22 November 2014), the Western part (Uvs Province; 14–21 July 2015), the Northern part (Khuvsgul Province; 19–25 July 2015), the Southern part (Umnugovi Province; 4–8 August 2016) and the Eastern part (Khentii Province; 9–12 August 2016). *H. pylori* from consecutive gastric cancer patients was collected from the National Cancer Center Hospital (Ulaanbaatar City; October 2015–August 2016). Written informed consent was obtained from all participants, and the ethical permission was approved by the Mongolian Ministry of Health (accepted number N3, 2015), Mongolian National University of Medical Sciences (N13-02/1A, 2013), and Oita University Faculty of Medicine (Yufu, Japan) (P-12-10, 2013). We obtained four biopsies (three for histology and one for culturing *H. pylori*) from all patients.

### 4.2. Histologic Diagnosis for Gastritis and Gastric Cancer

We followed the guidelines set by the American Society for Gastrointestinal Endoscopy for gastric mucosal sampling for histological diagnosis [27]; one from the antrum (approximately 2 cm from the pyloric ring in the greater curvature), one from the greater curvature of the corpus (8–10 cm from the esophagogastric junction) and one from the lesser curvature of angulus. We followed the same guidelines for gastric cancer patients, but the angulus specimen was not obtained due to ethical committee regulations. If suspected gastric cancer existed, at least one more specimen was taken for histological diagnosis. We excluded cases from further analysis if only small tissues or poor histological preparations were obtained. All gastric biopsy specimens were fixed in 10% buffered formalin and embedded in paraffin. Serial sections were stained with hematoxylin and eosin and the May–Giemsa stain. For suspected gastric cancer cases, the stained slides were described using Lauren’s classification and the Japanese classification for gastric carcinoma [28,29]. 

For non-malignant mucosal status, the acute inflammation (polymorphonuclear neutrophil infiltration), chronic inflammation (mononuclear cell infiltration), atrophy, IM, and bacterial density were classified into four grades according to the updated Sydney system: 0, “normal”; 1, “mild”; 2, “moderate”; and 3, marked” [14]. Samples with a grade of 1 or higher were considered as positive status. 

### 4.3. H. pylori Isolation and Sequencing

For *H. pylori* culture, antral biopsy specimens were homogenized in sterile normal saline solution and cultivated in a commercially available selective plate (Nissui Pharmaceutical Co. Ltd, Tokyo, Japan). The plates were incubated for up to 7–10 days at 37 °C under microaerophilic conditions (10% O2, 5% CO2, and 85% N2). *H. pylori* were identified based on colony morphology, Gram staining, and positive reactions for oxidase, catalase and urease tests. *H. pylori* DNA was extracted from multiple colonies using the QIAamp DNA Mini Kit (QIAGEN, Valencia, CA, USA) using the manufacturer’s directions. The cagA genotype was determined using polymerase chain reaction (PCR) amplification and direct sequencing of a conserved region of cagA as described previously (East-Asian type or Western-type) [16]. The cagA-negative status was confirmed using the cag empty site PCR as described previously [16]. A phylogenetic tree of *H. pylori* amino acid sequences of the C-terminal repeat region of CagA was constructed using MEGA7 (http://69.36.184.213/megabeta.php) and confirmed by visual inspection to determine the subtype (e.g., ABC, ABD). Additionally, vacA genotyping (s1 or s2, and m1 or m2) was performed as described previously [8,17]. 

### 4.4. Statistical Analysis

Statistical significance of the qualitative differences was calculated using the chi-square and Fisher’s exact test, and quantitative variables were tested using the Mann–Whitney test. In univariate analysis, all factors with *p* values less than 0.25 were involved in multivariate analysis. All statistical analyses were performed using the IBM SPSS Version 22.0 (IBM Corp. Armonk, NY, USA) software.

## 5. Conclusions

In summary, the current concept of different CagA types might not explain the geographic differences in the high incidence of gastric cancer, especially in Mongolia. 

## Figures and Tables

**Figure 1 cancers-11-00725-f001:**
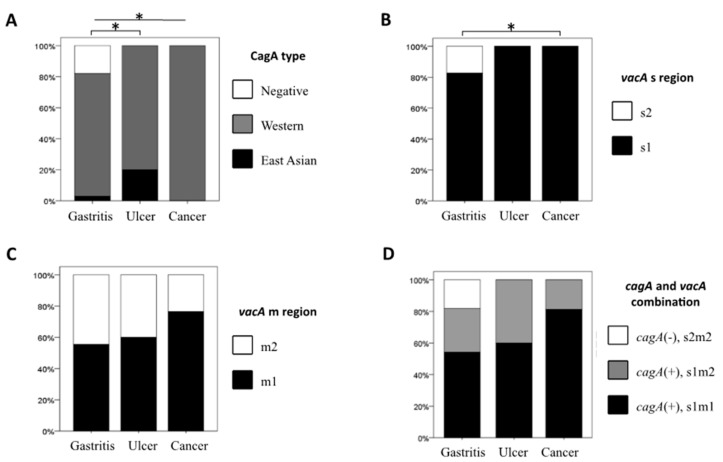
CagA and *vacA* genotyping based on diseases. The distribution of CagA types and *vacA* genotyping according to disease groups are shown. (**A**) CagA type, (**B**) *vacA* s region genotypes, (**C**) *vacA* m region genotypes and (**D**) combination of CagA and *vacA* typing. * *p*-value < 0.05.

**Figure 2 cancers-11-00725-f002:**
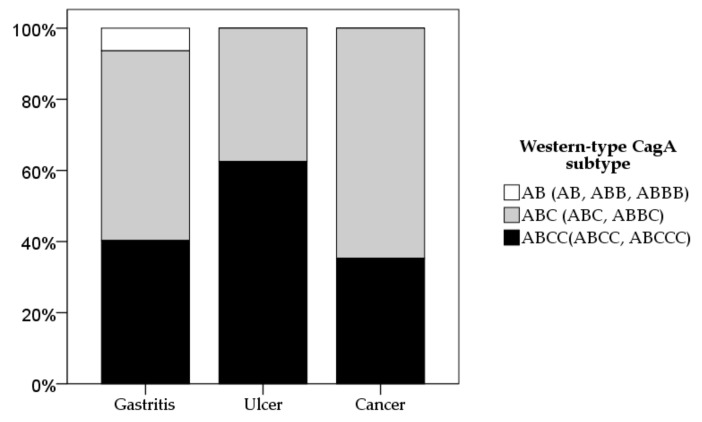
Western-type CagA subtyping based on diseases. The distribution of Western-type CagA subtypes based on disease groups are shown.

**Figure 3 cancers-11-00725-f003:**
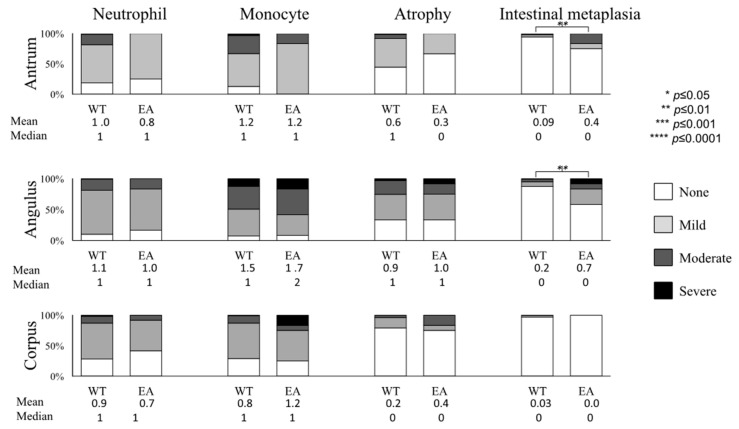
The Western-type CagA versus East-Asian type CagA based on histological status. The Western-type CagA versus East-Asian-type CagA based on histological status. The distribution and the mean and median values of histological features of gastritis according to CagA typing are shown. WT: Western-type CagA, EA: East-Asian-type CagA.

**Figure 4 cancers-11-00725-f004:**
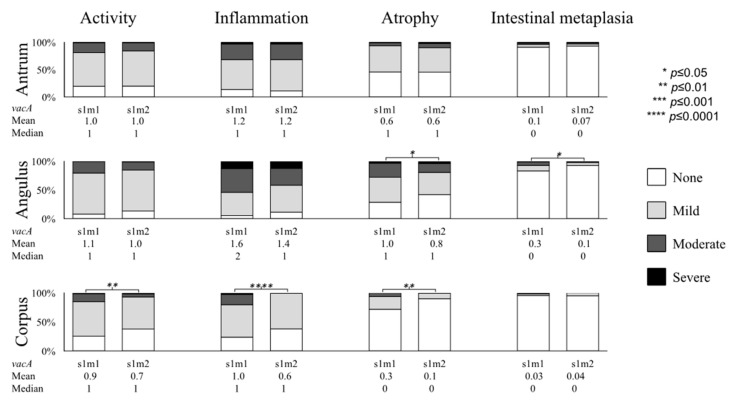
The *vacA* s1m1 versus s1m2 subtypes based on histological status. The distribution and the mean values of histological features of gastritis based on *vacA* genotyping are shown.

**Table 1 cancers-11-00725-t001:** The CagA types based on diseases.

CagA Types		Gastric Cancer *n* = 17 (%)	Gastritis *n* = 341 (%)	Peptic Ulcer *n* = 10 (%)	Total *n* = 368 (%)
Negative (−)		0 (0)	62 (18.2)	0 (0)	62 (16.8)
Western	^a^ AB	0 (0)	17 (5.0)	0 (0)	17 (4.7)
	^b^ ABC	11 (64.7)	142 (41.6)	3 (30)	156 (42.4)
	^c^ ABC*	6 (35.3)	109 (32.0)	5 (50)	120 (32.6)
East-Asian	^d^ ABD	0 (0)	10 (2.9)	2(20)	12 (3.2)
Hybrid	ABDC	0 (0)	1 (0.3)	0 (0)	1 (0.3)

^a^ AB (AB *n* = 15, ABB *n* = 1, ABBB *n* = 1) type was defined as Western-type CagA because the sequence of the B segment was mostly identical with the Western-type B segment: (KVNKKKAGQAANPEEPIYTQVAKKVNAKIDRLNQIASGLGGVGQAAGFPLKRHDKVDDLSKVG). ^b^ ABC (ABBC *n* = 3, ABC *n* = 153), ^c^ ABC* (ABCC *n* = 114, ABCCC *n* = 6), ^d^ ABD (ABD *n* = 9, ABBD *n* = 3).

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
