# Peer review of "Western-Type Helicobacter pylori CagA are the Most Frequent Type in Mongolian Patients"

_cancers, 2019, doi:10.3390/cancers11050725_

Round 1
Reviewer 1 Report
The aim of the article “Helicobacter pylori CagA types cannot explain the high gastric cancer incidence in Mongolia” was to determine the relationship between the virulence profile of H. pylori strains isolated from Mongolian patients and the incidence and severity of gastritis, gastric ulcers and gastric cancers. The subject of the manuscript is important, because existence in the geographically-dependent course of many diseases, also these caused by H. pylori, is increasingly noticed.
The article is written correctly, in a comprehensible way, with the occurrence of some small mistakes. Examples of suggested corrections:
1Introduction à 1. Introduction
“Helicobacter pylori is the infectious disease that causes chronic atrophic gastritis, peptic ulcer disease (PUD), gastric cancer, mucosa-associated lymphoid tissue (MALT) lymphoma and even extra gastric diseases such as idiopathic thrombocytopenic purpura” à e.g. Helicobacter pylori is a Gram-negative bacterium responsible for chronic atrophic gastritis, peptic ulcer disease (PUD), gastric cancer, mucosa-associated lymphoid tissue (MALT) lymphoma and even extra gastric diseases such as idiopathic thrombocytopenic purpura (H. pylori is a bacterium, not a disease !) [Introduction]
“Approximately 90% of gastric cancer …” à Approximately 90% of gastric cancers [Introduction]
“Importantly, all cagAnegative strains were isolated …” à Importantly, all cagA-negative strains were isolated [Results]
“… all strains from gastric cancer patients was Western-type CagA” à all strains from gastric cancer patients were Western-type CagA [Results]
“2.3. Gastric mucosal status with respect to H. pylori virulence factors” à 2.4. Gastric mucosal status with respect to H. pylori virulence factors [Results]
“Finally, we previously reported that the majority of gastric cancer in Mongolia was located in the upper part of stomach” à Finally, we previously reported that the majority of gastric cancers in Mongolia were located in the upper part of stomach [Discussion]
“Therefore, our current data support the current consensus …” à e.g. “Therefore, our present data support the current consensus OR Therefore, our data support the current consensus [Discussion]
I am asking you to edit the references from the number 20 (the whole has completely jumped to its place) [References]
In addition, it seems to me that it would be beneficial to add the statistics results to the “2.3. CagA and vacA genotyping according to diseases“ part in the Results.
Author Response
Responses for Reviewers comments
Responses for Reviewer 1
Q1: “Helicobacter pylori is the infectious disease that causes chronic atrophic gastritis, peptic ulcer disease (PUD), gastric cancer, mucosa-associated lymphoid tissue (MALT) lymphoma and even extra gastric diseases such as idiopathic thrombocytopenic purpura” à e.g. Helicobacter pylori is a Gram-negative bacterium responsible for chronic atrophic gastritis, peptic ulcer disease (PUD), gastric cancer, mucosa-associated lymphoid tissue (MALT) lymphoma and even extra gastric diseases such as idiopathic thrombocytopenic purpura (H. pylori is a bacterium, not a disease !)
Response: Thank you very much for your important correction. We corrected the sentence throughout the text carefully, and we also changed the first paragraph of introduction.
Q2: “Approximately 90% of gastric cancer …” Approximately 90% of gastric cancers [Introduction]
Q3: “Importantly, all cagA negative strains were isolated …” Importantly, all cagA-negative strains were isolated [Results]
Q4: “… all strains from gastric cancer patients was Western-type CagA” à all strains from gastric cancer patients were Western-type CagA [Results]
Q5: “2.3. Gastric mucosal status with respect to H. pylori virulence factors” à 2.4. Gastric mucosal status with respect to H. pylori virulence factors [Results]
Q6: “Finally, we previously reported that the majority of gastric cancer in Mongolia was located in the upper part of stomach” à Finally, we previously reported that the majority of gastric cancers in Mongolia were located in the upper part of stomach [Discussion]
Q7: “Therefore, our current data support the current consensus …” à e.g. “Therefore, our present data support the current consensus OR Therefore, our data support the current consensus [Discussion]
Response: We corrected these mistakes, and our manuscript was checked by a professional English editing service (Honyaku Center Inc. http://www.honyakucenter.jp/).
Q8: I am asking you to edit the references from the number 20 (the whole has completely jumped to its place) [References]
Response: We re-checked our references.
Q9: In addition, it seems to me that it would be beneficial to add the statistics results to the “2.3. CagA and vacA genotyping according to diseases“ part in the Results.
Response: We added p values in the section “CagA and vacA genotyping based on diseases.”

Reviewer 2 Report
In this paper the authors present the results of a national wide study aiming to evaluate the CagA status among 26 Mongolian population.
The section Introduction is too long. It should be shortened. In particular, the part from line 72 to 82.
The order of all sections should be revised. In particular, the section Materials and Methods is after that of discussion
Some reference should be updated. For example the number 1 (dated 2006) could be replaced by a more erecent review puiblished by the group of Prof. Mégraud (Pellicano et al. Panminerva Med 2016;58:304-17).
Author Response
Responses for Reviewers comments
Responses for Reviewer 2
Q1: The section Introduction is too long. It should be shortened. In particular, the part from line 72 to 82.
Response: Thank you very much for the suggestion. We shortened the introduction.
Q2: The order of all sections should be revised. In particular, the section Materials and Methods is after that of discussion
Response: We followed the template format recommended by the Cancers journal.
Q3: Some reference should be updated. For example the number 1 (dated 2006) could be replaced by a more recent review published by the group of Prof. Mégraud (Pellicano et al. Panminerva Med 2016;58:304-17).
Response: We tried to use recent references. Regarding the sentence related to original ref. 1, we changed the sentence according to the comments from Reviewer 3.

Reviewer 3 Report
The present cross sectional study aims to evaluate the CagA status in H. pylori clinical isolates from Mogolian patients, including chronic gastritis, peptic ulcer disease and gastric cancer. Although, this study is one of the first evaluating cagA-status in Mongolia, a country that exhibit one of the highest incidence and mortality of gastric cancer in the world, the study presents major drawbacks and lacks quality.
Major comments:
1. The study design is a cross sectional study, which by nature should be a descriptive study limited to a certain period of time (November 2014 – August 2016) that presents the current state in frequencies and/or prevalence. In contrast, the current study was analysed in a view of a case control scenario. As proof, the Title of the manuscript is an overstatement: “Helicobacter pylori CagA types cannot explain the high gastric cancer incidence in Mongolia”. Authors only studied 17 gastric cancers (~5% of the total individuals) all harbouring the Western-type CagA. Moreover, the current population size is too small in order to make associations between risk factors and gastric cancer incidence. Therefore, more soft language should be used in the title, such as “Helicobacter pylori CagA of the Western-type are the most frequent type among Mongolian patients”.
2. The manuscript contains several scientific inaccuracies:
2.1. Helicobacter pylori is NOT an infectious disease! In fact, is an infectious agent.
2.2. Helicobacter pylori infection causes chronic gastritis (i.e., induces chronic inflammation in the gastric mucosa) which depending on the pattern of inflammation throughout the stomach it can causes peptic ulcers or it can lead to the development of premalignant lesions. The development of gastric cancer occurs in the context of chronic atrophic gastritis, a consequence of the infection and not a direct effect as the authors mention in the text.
3. The statistical analyses are incomplete. The authors only refer that “statistical significance of qualitative differences was calculated using chi-square and Fisher’s exact test”. First, it should be defined each variable were considered as “qualitative”. Secondly, the analysis of quantitative variables should be mentioned. Quantitative variables such as scores of Neutrophils, Monocytes, Atrophy, Intestinal Metaplasia should be analysed with non-parametric tests (compares median across groups) and not with chi-square and Fisher’s exact test (compares proportion across groups).
4. The section “5. Conclusions”, does not make any sense. The authors try to make a brief summary of the manuscript, which are badly understandable, instead of the producing clear conclusions based on their data.
5. The manuscript needs urgent English grammar corrections. Most of the paragraphs are difficult to understand due to lack of concordance in verbal tense and clarity in the terms used.
Minor comments:
1. Regarding the CagA-types, since the western-type are the most frequent, the authors should look for an association between the number of EPIYA C motifs and the different clinical settings (Gastritis vs Gastric Cancer or vs Peptic Ulcer).
2. The terms “histological background” and “histological score” should be substituted by “histological status” and “histological features of gastritis” respectively.
3. The figure legends are poorly written and incomplete. In fact there are two different entries for each Figure legend, one above and other below the Figure.
4. In Table 1, the sum of the percentages in the column “Gastritis” is equal to 105% and should be 100%.
5. Patient demographics. Patient’s features such as age and gender should be presented in a table in regard of their clinical setting.
6. When referring to names of genes, these should be written in italic.
Author Response
Responses for Reviewers comments
Responses for Reviewer 3
Q1: The study design is a cross sectional study, which by nature should be a descriptive study limited to a certain period of time (November 2014 – August 2016) that presents the current state in frequencies and/or prevalence. In contrast, the current study was analysed in a view of a case control scenario. As proof, the Title of the manuscript is an overstatement: “Helicobacter pylori CagA types cannot explain the high gastric cancer incidence in Mongolia”. Authors only studied 17 gastric cancers (~5% of the total individuals) all harbouring the Western-type CagA. Moreover, the current population size is too small in order to make associations between risk factors and gastric cancer incidence. Therefore, more soft language should be used in the title, such as “Helicobacter pylori CagA of the Western-type are the most frequent type among Mongolian patients”.
Response: Thank you very much for the important correction. We changed the title according to your suggestion and mentioned the limitation of our study in the Discussion. In addition, we softened the importance of CagA types in Discussion. For example, we used “association” instead of “induce”.
Q2: The manuscript contains several scientific inaccuracies:
2.1. Helicobacter pylori is NOT an infectious disease! In fact, is an infectious agent.
Response: Thank you very much for your important correction. We corrected the sentence in the Introduction.
2.2. Helicobacter pylori infection causes chronic gastritis (i.e., induces chronic inflammation in the gastric mucosa) which depending on the pattern of inflammation throughout the stomach it can causes peptic ulcers or it can lead to the development of premalignant lesions. The development of gastric cancer occurs in the context of chronic atrophic gastritis, a consequence of the infection and not a direct effect as the authors mention in the text.
Response: This is really an important and useful comment, and we mentioned the concept in the first lines of the Introduction.
Q3. The statistical analyses are incomplete. The authors only refer that “statistical significance of qualitative differences was calculated using chi-square and Fisher’s exact test”. First, it should be defined each variable were considered as “qualitative”. Secondly, the analysis of quantitative variables should be mentioned. Quantitative variables such as scores of Neutrophils, Monocytes, Atrophy, Intestinal Metaplasia should be analysed with non-parametric tests (compares median across groups) and not with chi-square and Fisher’s exact test (compares proportion across groups).
Response: We used non-parametric tests in the original version. To make it easier for readers to understand, we also used “mean” for the Sydney system score (please see Figure 3 and 4; we used both “mean” and “median”), but the analyses were done using the Mann-Whitney test. We also added the Mann-Whitney test for quantitative variables in the methods.
Q4. The section “5. Conclusions”, does not make any sense. The authors try to make a brief summary of the manuscript, which are badly understandable, instead of the producing clear conclusions based on their data.
Response: Thank you very much for your critical comment. We revised the discussion and conclusion to be more understandable.
Q5. The manuscript needs urgent English grammar corrections. Most of the paragraphs are difficult to understand due to lack of concordance in verbal tense and clarity in the terms used.
Response: The manuscript was revised by a professional English editing service (Honyaku Center Inc. http://www.honyakucenter.jp/).
Q6. Regarding the CagA-types, since the western-type are the most frequent, the authors should look for an association between the number of EPIYA C motifs and the different clinical settings (Gastritis vs Gastric Cancer or vs Peptic Ulcer).
Response: We made a new figure (Figure 2) for the Western-type CagA typing according to diseases.
Q7. The terms “histological background” and “histological score” should be substituted by “histological status” and “histological features of gastritis” respectively.
Response: Thank you very much for your important comment. We corrected the terminology throughout the main text and supplementary files.
Q8. The figure legends are poorly written and incomplete. In fact there are two different entries for each Figure legend, one above and other below the Figure.
Response: We revised each figure legend.
Q9. In Table 1, the sum of the percentages in the column “Gastritis” is equal to 105% and should be 100%.
Response: We apologize for the mistake. We corrected the percentage in Table 1.
Q10. Patient demographics. Patient’s features such as age and gender should be presented in a table in regard of their clinical setting.
Response: We made the new table in supplementary file (Supplementary table 1).
Q11. When referring to names of genes, these should be written in italic.
Response: We corrected it as you suggested.

Round 2
Reviewer 3 Report
Please include references for the following sentece (lines 201-202):
"This consensus is based on several previous reports."